# Household medication safety practices during the COVID-19 pandemic: a descriptive qualitative study protocol

Tamasine C Grimes ![ORCID],[1] Sara Garfield,[2,3] Dervla Kelly ![ORCID],[4] Joan Cahill,[5] Sam Cromie,[5] Carly Wheeler,[3] Bryony Dean Franklin[2,3]

¹School of Pharmacy and Pharmaceutical Sciences, Trinity College Dublin, Dublin, Ireland
²UCL School of Pharmacy, University College London, London, UK
³Centre for Medication Safety and Service Quality, Imperial College Healthcare NHS Trust, London, UK
⁴School of Medicine, University of Limerick, Limerick, Ireland
⁵Centre for Innovative Human Systems (CIHS), School of Psychology, Trinity College Dublin, Dublin, Ireland

**Correspondence to**
Professor Bryony Dean Franklin;
bryony.deanfranklin@ucl.ac.uk

## ABSTRACT

**Introduction** Those who are staying at home and reducing contact with other people during the COVID-19 pandemic are likely to be at greater risk of medication-related problems than the general population. This study aims to explore household medication practices by and for this population, identify practices that benefit or jeopardise medication safety and develop best practice guidance about household medication safety practices during a pandemic, grounded in individual experiences.

**Methods and analysis** This is a descriptive qualitative study using semistructured interviews, by telephone or video call. People who have been advised to 'cocoon'/'shield' and/or are aged 70 years or over and using at least one long-term medication, or their caregivers, will be eligible for inclusion. We will recruit 100 patient/carer participants: 50 from the UK and 50 from Ireland. Recruitment will be supported by our patient and public involvement (PPI) partners, personal networks and social media. Individual participant consent will be sought, and interviews audio/video recorded and/or detailed notes made. A constructivist interpretivist approach to data analysis will involve use of the constant comparative method to organise the data, along with inductive analysis. From this, we will iteratively develop best practice guidance about household medication safety practices during a pandemic from the patient's/carer's perspective.

**Ethics and dissemination** This study has Trinity College Dublin, University of Limerick and University College London ethics approvals. We plan to disseminate our findings via presentations at relevant patient/public, professional, academic and scientific meetings, and for publication in peer-reviewed journals. We will create a list of helpful strategies that participants have reported and share this with participants, PPI partners and on social media.

## INTRODUCTION

### Problem statement and knowledge gap

Medication use is the most common intervention in healthcare. In the past decade, prior to the COVID-19 pandemic, more than 8% of all adult emergency hospital admissions and 25% of those for older adults (65 years and over) have been related to medication use, with increasing age and higher disease burden being key risk factors.[1 2] In 2020, the world's population is

### Strengths and limitations of this study

► This study will provide novel insights into the household medication safety practices of those who are cocooning/shielding during a pandemic, leading to the creation of best practice guidance about household medication safety practices during a pandemic from the patient's/carer's perspective.

► Data collection methods (telephone and video call) will enable us to reach vulnerable people who are staying at home during the pandemic, while maintaining public health guidance for avoiding face-to-face contact.

► Because interviews are limited to the English language, the perspectives of people who do not speak English confidently are likely to be under-represented. In addition, the perspectives of those without telephone or internet access will not be obtained.

► It is possible that theoretical saturation may not be reached due to resource constraints, which limit the capacity for recruitment to fifty participants per country.

predicted to use 4.5 trillion medication doses, up 24% from 2015, and more than 50% will consume more than one dose per day.[3] A recent American study identified the annual cost of medication-related morbidity and mortality, consequent to non-optimised medication use, as €528 billion, equivalent to 16% of healthcare expenditure.[4] As a result of such problems, the current WHO Global Patient Safety Challenge seeks to reduce the burden of serious, avoidable medication-related harm by 50% within 5 years from 2017.[5]

Optimal medication use requires input from multiple stakeholders, including consumers, patients and carers, operating within complex adaptive systems.[6] Evidence suggests that contribution by patients, informal carers and family members ('lay involvement') is a potentially important modifiable factor in optimising medication outcomes and providing system resilience.[7–10] Examples of lay involvement

include patients and family members routinely taking on medication management roles,[9 11–13] often of complex medication regimens and in challenging contexts, such as in dementia,[14–16] heart failure,[17] chronic obstructive pulmonary disease,[18] cancer[19] and home palliative care.[20] However, there are known hazards in lay medication practices, including errors in medication administration,[9] sharing or borrowing prescribed medication,[21 22] unsafe or unsuitable storage[23 24] or disposal,[19] hoarding,[17 25] stockpiling[17] and non-adherence.[26] Therefore, there is much to address before the full potential of lay involvement in safe household medication use can be realised.

The literature suggests that medication use poses a significant body of medication-related 'work' for lay users.[11 17 18 27] Specifically, this suggests that patients themselves manage the majority of their medication work, but that a proportion is managed by others who are members of the person's social network.[11] Explorations of the nature of lay work involved in medication practices, whether for self or others, have identified a range of strands to this work. The work includes planning and coordination of medication taking and adherence, emplacement of medication in the home or other personal spaces, tracking of medication supplies, monitoring of effects of medication use, monitoring for and management of one's own or others' errors, management of information from professional or non-professional and personal sources and interactions between all of these.[11 17 18 27–29] Several studies identify that lay medication work involves an emotional burden, often associated with a response to the need to use or rely on medication, the toll of the work involved and the response to one's own or another's efforts around medication adherence.[11 17 18 27 28] Medication use has been identified as a tool to facilitate a person to (re)gain control in the personal context of a relatively less controlled or chaotic experience of illness.[11 17] There is evidence that medication taking and medication work are highly personalised and complex phenomena that may be contingent on, and contextualised by one's situation, environment, processes and network.[11 17 30] Examples of such complexities include the medication user's individualised routines and strategies, the nature of medication(s) involved, the range of people involved or the individual's illness burden.[11] Studies have identified that medication work comprises sociotechnical efforts that are often complex,[17] and that lay people develop, adapt and rely on a variety of strategies to achieve personal goals associated with medication use.[11 17 18 29 30]

In their study of older adults with heart failure, Mickelson *et al* argue that medication management is the most common performed self-care behaviour in this cohort.[17] Cheraghi-Sohi *et al*[11] developed a framework of 'medication work' using social network analysis to map the work undertaken by people with long-term conditions (LTCs) and their wider social networks, that is, people identified in some way to be of relevance to the person's medication management.[11] Schafheutle *et al* subsequently applied and adapted this framework to people experiencing chronic obstructive pulmonary disease.[18] Both Cheraghi-Sohi *et al*'s and Schafheutle *et al*'s studies were based on interviews undertaken with people with

LTCs and therefore provide the patient's/carer's perspective, which is also central to the present work.[11 18] However, this body of evidence has been generated outside of the context and constraints of a pandemic and studies have not examined populations staying at home for long periods.

During the COVID-19 pandemic, people who are self-isolating and housebound for long periods of time, whether due to age or 'extreme medical vulnerability', are potentially at increased risk of medication-related problems or harm. These may include medication errors, adverse drug reactions, adverse drug events and medication non-adherence. The key clinical issues with the potential to affect medication safety for those who are self-isolating for long periods of time during the COVID-19 pandemic are: (1) disrupted routine healthcare services and supply chains; (2) altered household mobility, well-being and support structures; (3) reluctance to attend healthcare or restrictions on attending healthcare and (4) misinformation about medications reported to affect the risk or severity of COVID-19 infection. These create additional challenges for standard medication safety, at a time when prevention of avoidable iatrogenic harm is particularly important.

### The present need for research

Relatively little is known about medication safety or changes to household medication practices during a pandemic. Limited evidence on medication use during previous pandemics identifies reduced health-seeking behaviour from clinics or hospitals, increased self-management of common or chronic ailments at home, increased self-medication, stockpiling antiviral medication and increased attendance at emergency departments for antiviral related adverse effects.[31–34] Disruption of household mobility, well-being and social network membership is likely for people who are staying at home and self-isolating during the COVID-19 pandemic. This disruption therefore has the potential to affect medication work and practices, and to alter routine defences or hazards, thereby affecting medication safety.

Previous studies have generally classified drug-related problems from a healthcare professional's perspective, rather than that of the person taking the medication or their social network; drug-related problems are also generally identified in the context of medication review, rather than routine household settings.[35] The patient's perspective of household medication use is central to the present study. The patient-centred medication safety framework is grounded in the patient's perspective of matters that they consider important to the management of medication safety incidents in primary care.[8] However, it does not address the broader concept of household medication safety practices, which is the focus of the current study.

Evidence supports the combined use of Safety-I and Safety-II approaches to identify when, where, how and why things go wrong (Safety-I) or right (Safety-II).[36] Medication practices occur within complex sociotechnical systems where people (agents and actors), artefacts, information and knowledge flows, technologies, environments and individual or collective behaviours meet in a dynamic and unpredictable

way.[6 17 37 38] A Safety-I/Safety-II perspective has previously been used to understand and create frameworks of hazards and defences associated with routine medication practices in the community,[8] hospital[38] and domiciliary settings,[12 39] involving medication safety incidents, intravenous infusion practices, informal carer medication administration and domiciliary medication safety among caregivers of chronically ill children. Mickelson *et al* used a systems engineering framework to support their analysis of medication management strategies by older patients with heart failure.[17] We are not aware of any study that has employed Safety-I/Safety-II approaches to support the analysis of household medication safety practices among either the general adult population or those who are housebound, or in the context of a pandemic.

This study therefore seeks to explore household medication practices for people who are staying at home and reducing contact with others as much as possible during the COVID-19 pandemic. It also aims to create best practice guidance about household medication safety practices during a pandemic from the patient's/carer's perspective, which can be used to enable patient and carer agency in safe medication practices during the COVID-19 pandemic and beyond.

Specific objectives are to:
► Explore experiences of patients (or their carers) around household medication safety practices during the pandemic.
► Identify practices and situations that may benefit or jeopardise medication safety practices during the pandemic.
► Create best practice guidance about household medication safety practices during a pandemic from the patient/carer perspective.

## METHODS AND ANALYSIS
### Researcher characteristics and reflexivity
The research team is composed of research pharmacists working in the academic setting (TCG, DK), and hospital and academic settings (BDF, SG), an organisational psychologist (SC) and senior researchers with expertise in human factors, well-being management and older adult enablement (JC) and health services research (CW). Patient and public involvement (PPI) partners will also be involved in data analysis.

### Context
We will conduct the study in the four devolved nations of the UK, and Ireland. All have implemented public health guidance to protect those who are extremely medically vulnerable during the COVID-19 pandemic. This includes advice for such people to stay at home as much as possible and reduce contact with other people, referred to as 'cocooning' in Ireland and 'shielding' in the UK.[40 41]

### Design
Qualitative exploration using semistructured interviews. Qualitative methods are central to exploring medication practices and medication safety, and have been used for this purpose in both domiciliary and hospital settings.[8 12 38 39]

### Definition of household medication safety practices
We define household medication safety practices as any activities by patients, their caregivers or members of their social networks related to obtaining supplies, deciding whether or not to use medicines, storing, consuming or administering medication, monitoring and responding to signs of therapeutic or adverse effects. We include prescribed, over-the-counter, complementary and alternative medication.

### Participants
#### Inclusion criteria
Adults will be eligible to participate if they experience a medical vulnerability listed on the government public health advice as requiring protection against COVID-19 infection and have been advised to 'shield'[40] or 'cocoon'[41] during the COVID-19 pandemic, and/or are aged 70 years or more, and they use at least one long-term medication. Carers who assist in the medication management of an adult who fulfils these two criteria are also eligible to take part.

#### Exclusion criteria
► Children under 18 years of age.
► People too unwell or otherwise unable to consent to interview.
► People not speaking English confidently.
► People without access to a telephone or a device with internet connection.

### Sampling strategy
Up to 100 adult research participants will be selected using convenience sampling, to include patients and carers living in the UK or Ireland. Our protocol does not include a screening procedure to assess the demographic or clinical characteristics of prospective research participants prior to study recruitment. However, following recruitment, we will create a matrix of brief background demographic characteristics of those previously interviewed to support iterative, focused recruitment of a diverse sample. We aim for the study population to represent people with a range of ages, genders, ethnicities, geographical areas, number of medicines and living alone or with others and those who are patients or carers.

### Recruitment
Research participants will be recruited through patient or carer advocacy groups or charities, by engaging with our PPI partners, personal or professional networks and through social media. Our focused recruitment will also support us to recruit a diverse sample of people with differing at-home medication needs and roles. For example, we will do focused recruitment of those in carer roles, by engaging with our PPI partners in carer advocacy groups. We will do focused recruitment of those experiencing particular medical vulnerabilities by engaging with charities and PPI partners working in that context, for example, cystic fibrosis, arthritis, cancer, organ transplant and old age. Prospective participants will be encouraged to make contact with a member of the research team, by telephone or email. Consistent with ethical, data protection and confidentiality regulations, we will not ask

## Box 1  Topic guide

1. Before the coronavirus pandemic, how did you get on managing your medicines, or the medicines for the person(s) to whom you provide care?
2. Since the coronavirus pandemic started, how has the way you managed or used medicines changed?
3. Have you experienced any difficulty using or managing medication during the pandemic?
4. What do you think would make it easier or better to manage medications at home during the pandemic?
5. What would you advise others to do in your situation?
6. Do you have anything else you would like to add?
7. Background and demographic information.

third parties to pass prospective participants' personal details to the research team.

### Data collection

We will use semistructured interviews conducted by telephone or video call. Data will be collected from June through October 2020. Patients will be invited to express their preferred mode of data collection. We aim to interview 50 participants in Ireland and 50 in the UK. With the participant's consent, interviews will be audio or audio–video recorded and/or the researcher will take detailed notes.

### Interview

A member of the research team will conduct the interview at a pre-agreed date and time. The interview will explore medication use by the patient participant or for the person(s) to whom a carer participant provides care. This interview will occur remotely using the participant's preferred method of telephone or video call. At no stage will the researcher and participant meet in person. Interviews are expected to last 15–45 min.

The proposed topic guide (box 1) was developed using a priori principles of routine lay medication use, applied in the context of the COVID-19 pandemic. Interviewers will use additional prompts and follow-up questions as needed.

### Data processing

Interview recordings will be transcribed verbatim (either automated or using a third-party service) and merged with the researcher notes. Any identifiable information will be redacted from the transcription. The recording will be permanently deleted at a time consistent with the ethics approval and data protection regulations in each country. The transcribed files will be imported into NVivo to support data analysis. We will restrict access to the identifiable interview files to the research team at each site (Dublin, Limerick or London). Following irreversible anonymisation, the data collected by the researcher(s) at each university will be processed and analysed at that university and may be transferred among them at the later stages of analysis to support comparison in line with ethical approvals.

### Analysis

A constructivist–interpretive analysis approach will be used. The constant comparative technique will be employed to systematically organise, compare and understand similarities and differences in the data.[42] Data analysis will occur in parallel to data collection. Inductive analysis will be used to identify the emergent medication practices, strategies, hazards and defences. Sociotechnical assessment approaches using a Safety-I/Safety-II perspective[36] and Schafheutle *et al*'s medication work framework[18] will serve as sensitising concepts[43] to explore the data, in which their application will be used to inspire insights appropriate to the data rather than being used as a rigid coding framework. This will support the iterative development of our best practice guidance about household medication safety practices relating to those who are staying at home for long periods during a pandemic.

### Research rigour, trustworthiness and credibility

We used the Standards for Reporting Qualitative Research recommendations to guide this protocol development.[44] In Ireland, research participants will be offered access to the recorded interview file, researcher notes or transcription. Member checks of these files are welcome but are not required. Each interviewer will quality assure their own interviews for transcription accuracy. A proportion of interviews will be coded independently by at least two members of the research team, followed by a comparison of and discussion of coding as a means to quality assure the coding process. Publication of this protocol is our first step in documenting and creating an audit trail of decisions made throughout the project and we will continue to capture these during data collection, analysis and reporting.

### Patient and public involvement

PPI will be facilitated through personal and professional networks, contacts within patient advocacy groups, charities and professional organisations. PPI commenced during the development phase by partners, including patients, validating our research aim and reviewing all participant-facing materials including recruitment materials, participant information leaflet, consent form and interview topic guide. Research participant recruitment will be supported by our PPI partners in patient and carer advocacy groups and other networks. PPI will extend to consultation on dissemination and outreach activities and involvement with data analysis and interpretation, using models employed previously.[45 46] All PPI will occur remotely, compliant with current public health advice for avoiding face-to-face contact. Our PPI partners will be offered the opportunity to coauthor the future papers describing our research findings.

### Ethical considerations

Approval for this study was granted by the Trinity College Dublin Faculty of Health Sciences Research Ethics Committee (reference COVID-19 2020502), University of Limerick Education and Health Sciences Research Ethics Committee (reference 2020_06_08_EHS) and University College London Research Ethics Committee (reference 18417.001).

The legal basis, in Ireland and the UK, for the processing of these data is scientific research in the public interest, however, we will invite informed consent from all participants prior to participation. In the context of the COVID-19 pandemic and infection control of this medically vulnerable group, verbal consent by audio-recorded telephone or video call, in cases where hard copy postage or electronic transmission poses a challenge, has been deemed acceptable by the ethics committees concerned. Compliant with contemporary data protection guidance for qualitative research during COVID-19, recordings of interviews will be permanently deleted as soon as specified in the ethics application at each site and transcripts will be irreversibly anonymised. As with all qualitative research, quotations from interviews will be used to illustrate the findings and, in all cases, will be anonymised and of a nature that cannot be traced back to any given person. If interviewers identify any pressing medication safety concerns, they will signpost the participant to relevant sources of support.

## Dissemination

We plan to present our work at suitable Irish, UK and international conferences and to publish at least one peer-reviewed research paper. We will produce plain English summaries of our work, which we will offer to send to participants, and we will engage with our PPI partners to optimise public outreach and research impact to the wider public. We will disseminate to healthcare professionals through professional bodies. We will produce a policy brief outlining key recommendations to support safer lay medication practices among those who are housebound in the context of a pandemic. Our research paper(s) will be made available open access and the summaries will be shared through our university repositories and social media channels, and through our PPI partners' social media channels.

## Impact

The proposed output from this research is a set of best practice guidance about household medication safety practices during a pandemic from the patient's/carer's perspective, relating to those who are staying at home and reducing contact with others as much as possible during the COVID-19 pandemic. We will create a list of helpful strategies that participants have reported and share this with participants, PPI partners and on social media, to use as a tool to enable agency and medication safety at home for people who are shielding, cocooning or equivalent during a pandemic.

The research priorities outlined in the WHO and Global Research Collaboration for Infectious Disease Preparedness joint research roadmap for the COVID-19 pandemic include: to rapidly identify secondary effects of the outbreak and deliver strategies to mitigate patient harm, and to rapidly involve communities in the design, delivery and dissemination of research.[47] The proposed study addresses both priorities by providing person-centred guidance, based on individual experiences, with significant PPI input. Enabling patients, carers and social network members to become agents in medication safety aligns with the WHO agreed global priorities for medication safety research.[48] Future work will explore the transferability of this guidance to a wider population of housebound and ambulatory people who are using medication long term, and outside of the context of a global pandemic. Given the forecasted increase in the global population of adults living well with age-related conditions over the coming decades,[49] an important consideration is to enable people to remain in their homes and to be agents in their own medication management and safety.[50 51]

**Acknowledgements** We are grateful for input from our patient and public involvement partners, including those who reviewed participant-facing materials: Alex Taylor; Marney Williams; Fran Husson; Jill Lloyd; John Norton; Mike Etkind; Dr Nikki Dunne, Research Officer, Family Carers Ireland; Corona Joyce, Senior Policy Officer, Age Action Ireland and Josephine Fogarty, Coordinator of Traveller Health, Midwest Community Healthcare, Health Service Executive, Ireland.

**Contributors** TCG, DK, BDF and SG led the study design. TCG led the manuscript drafting. All authors contributed to manuscript revision and approval.

**Funding** This work was supported by the Health Research Institute, University of Limerick, and the National Institute for Health Research Imperial Patient Safety Translational Research Centre (grant number SF003). This report presents independent research.

**Disclaimer** The views expressed in this publication are those of the authors, not necessarily those of the Health Research Institute, University of Limerick, the NHS, the National Institute for Health Research or the Department of Health and Social Care.

**Competing interests** None declared.

**Patient and public involvement** Patients and/or the public were involved in the design, or conduct, or reporting, or dissemination plans of this research. Refer to the Methods and analysis section for further details.

**Patient consent for publication** Not required.

**Provenance and peer review** Not commissioned; externally peer reviewed.

**ORCID iDs**
Tamasine C Grimes http://orcid.org/0000-0002-7154-3243
Dervla Kelly http://orcid.org/0000-0001-9836-5400

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
