## [Reviewer comments · BMJ Open]

ARTICLE DETAILS

TITLE (PROVISIONAL)	Household Medication Safety Practices during the Covid-19 Pandemic: A Descriptive Qualitative Study Protocol
AUTHORS	Grimes, Tamasine C.; Garfield, Sara; Kelly, Dervla; Cahill, Joan; Cromie, Sam; Wheeler, Carly; Dean Franklin, Bryony

VERSION 1 – REVIEW

REVIEWER	DK Theo Raynor University of Leeds, UK I am academic advisor to Luto Research which develops, refines and tests health information materials. I am European co-chair of the Bayer 'Patient Safety Communication Advisory Panel'.
REVIEW RETURNED	16-Sep-2020

GENERAL COMMENTS	This protocol is well written and comprehensive and addresses an important and current medication safety topic. Although the research team contains a good mix of academics, I note that there is no patient co-applicant. This would have been an appropriate addition to the current well developed PPI. The terminology in the title: 'household medication safety' made me immediately think of medicines kept in households in medicine cabinets - often referred to as 'household medicines'. Would it be better understood if described as 'Safe use of medicines in the home ...'? The abstract mentions a 'Safety I/II perspective' - many people will not be familiar with this concept. Although it is expanded upon in the main part of the protocol, a brief sentence of explanation is needed here. The terminology used is not consistent: 'lay-centred framework' vs 'patient-centred framework' vs 'person-centred'.. Line 24: 'Reluctance to attend healthcare' could be better described as "Reluctance to attend healthcare or restrictions on attending healthcare". Line 25: It is not clear how 'misinformation about medicines' is different during the Covid-19 pandemic than in normal times.
--

	On line 57 onwards, the description of household medication practices should also include 'accessing and acting on information about medicines'. This is also missing from the Topic Guide.
--	---

REVIEWER	Elizabeth Manias Deakin University Australia
REVIEW RETURNED	26-Sep-2020

GENERAL COMMENTS	General This protocol provides details about the development of an approach to comprehensively investigate household medication practices among adults during the Covid-19 pandemic. Overall, the paper is well-written. However, there are a number of areas that need to be considered to improve the quality of the protocol, to ensure transparency of the protocol, and to make it relatively easy for readers to follow. Abstract During the Covid-19 pandemic, there would be many people who are staying at home, and attempting to reduce contact with other people. Additional clarification is needed to identify these individuals. Clarification is needed in what is meant by 'as much as possible' in relation to individuals who are self-isolating. If keeping the concepts relating to the Safety I and II perspective in the abstract, these concepts should be briefly defined. If not, then these concepts should be removed from the abstract. Introduction There are a number of terms in the introduction that should be defined more specifically to aid understanding. Clarification is needed about what is meant by 'normal times.' Similarly, an explanation is needed of what is meant by 'lay involvement' in medication practices, as well as of 'medication performance' and of 'medication work.' The authors stated that evidence exists identifying that medication taking, and medication work, is a highly personalised and complex phenomenon that may be contingent upon, and contextualised by, one's situation, environment, processes and network. However, they provided little explanation of the complexities relating to influences in relation to what is already known from available evidence. It is not clear what the authors mean by 'other events of major public health concern,' since in the preceding part of the sentence, they referred to lack of details about medication safety during pandemics. The authors stated that previous studies describing drug-related problems have been undertaken from the healthcare professionals' perspective, rather than from the person taking the medication. Does this statement refer during the time of a pandemic or outside of a pandemic event? There have been numerous studies undertaken of the patients' perspective in relation to drug-related problems that consider the broader
---

	concepts of household medication practices, which the authors have not described or critiqued. The aims and the specific objectives of the study appear appropriate. Methods and analysis It is stated that patient and public involvement partners will be involved in data analysis. Clarification is needed about whether these individuals were involved in the development phases of the project. For inclusion criteria, clarification is needed about which adults comprise those individuals who experience a medical vulnerability as it could be assumed that many individuals could encompass this group. Similarly for the remaining two criteria, there are many adults who are on at least one long-term medication or who have assistance from another individual with medication management. While the authors intend to use a convenience sampling process, they will develop a matrix to document participants' backgrounds to ensure focused recruitment of a diverse sample. It would be helpful to know about the types of individuals who utilise patient and carer advocacy groups, engagement with public and patient involvement, personal networks and social media. It is important potential participants comprise a wide array of characteristics to ensure adequate representation. It is encouraging to know that the authors will also seek out recruitment from charity groups. The actual process of the recruitment has not been described, and further information is needed about this activity. How will the authors obtain the potential participants' contact details in order to arrange an interview? For data analysis, clarification is needed about how sociotechnical assessment approaches using Safety I and II and Schafheutle et al.'s medication work framework will be applied. For research rigour, the authors stated that a proportion of interviews will be quality assured by another member of the research team for validation of coding and analysis. Further explanation is needed about this process. Impact The authors seek to develop a person-centred medication practice safety framework for individuals staying at home and with reduced contact with others. From what is provided in the analysis section, which involves a constructivist-interpretive analysis approach, it is not clear how a person-centred medication practice safety framework will be developed. Further clarification is needed about the process of development of this framework and of the ways in which development of this framework will be influenced by other frameworks to be used during analysis.
--	---

REVIEWER	Dianne Goeman
	University of Newcastle, Australia
REVIEW RETURNED	27-Sep-2020

GENERAL COMMENTS	This protocol paper describes a study that will explore the household medication practices of people over 70 yrs of age in the UK and the Republic of Ireland who are confined to home during the Covid-19 pandemic. The purpose of the proposed study is to develop a person-centred framework (including a list of strategies) of safe household medication practices to follow during a pandemic. As the protocol paper is well written, meets all of the review checklist criteria and the proposed person-centred framework of safe medication practices is novel I recommend that the protocol paper be published.
--

VERSION 1 – AUTHOR RESPONSE

Reviewers' Comments to the Author	RESPONSES
Reviewer: 1 Reviewer Name: DK Theo Raynor Institution and Country: University of Leeds, UK Please state any competing interests or state 'None declared': I am academic advisor to Luto Research which develops, refines and tests health information materials. I am European co-chair of the Bayer 'Patient Safety Communication Advisory Panel'.	
This protocol is well written and comprehensive and addresses an important and current medication safety topic. Although the research team contains a good mix of academics, I note that there is no patient co-applicant. This would have been an appropriate addition to the current well developed PPI.	Many thanks for your constructive review and helpful comments on the article. Thank you for your consideration of the research team composition and patient contribution. We notice you mention co-applicant, and we are assuming that you mean co-author and are responding to your comment on that basis. PPI input to date has been highly valuable, as described in the protocol. Although our PPI partners did not meet the criteria for authorship of the study protocol, they are listed in the acknowledgements section, and have been offered the opportunity for co-authorship on the final paper describing our findings. We have now clarified this in the protocol manuscript.
The terminology in the title: 'household medication safety' made me immediately think of medicines kept in households in medicine cabinets - often referred to as 'household medicines'. Would it be better understood if described as 'Safe use of medicines in the home ...'?	Thank you for your suggestion and call for clarity in the title. We have revised this to "Household Medication Safety Practices during the Covid-19 Pandemic: A Descriptive Qualitative Study Protocol". We believe this captures the essence of safe use of

	medicines in the home and focusses on the practices rather than on the storage.
The abstract mentions a 'Safety I/II perspective' - many people will not be familiar with this concept. Although it is expanded upon in the main part of the protocol, a brief sentence of explanation is needed here.	Upon reflection, we have removed reference to this sensitising concept from the abstract and have introduced it in the main paper where we are better able to explain these concepts.
The terminology used is not consistent: 'lay-centred framework' vs 'patient-centred framework' vs 'personcentred'..	Thank you for this valuable observation. We have reviewed the manuscript to ensure consistent use of the term "person-centred" as this reflects our consideration of both patients and carers.
Line 24: 'Reluctance to attend healthcare' could be better described as "Reluctance to attend healthcare or restrictions on attending healthcare".	Again, thanks for this valuable suggestion. We agree and have revised accordingly.
Line 25: It is not clear how 'misinformation about medicines' is different during the Covid-19 pandemic than in normal times.	Thanks for prompting us to clarify this. We have changed this to "misinformation about medicines reported to affect the risks or severity of Covid-19 infection"
Reviewer: 2 Reviewer Name: Elizabeth Manias Institution and Country: Deakin University, Australia Please state any competing interests or state 'None declared': None declared.	RESPONSES
General This protocol provides details about the development of an approach to comprehensively investigate household medication practices among adults during the Covid-19 pandemic. Overall, the paper is well-written. However, there are a number of areas that need to be considered to improve the quality of the protocol, to ensure transparency of the protocol, and to make it relatively easy for readers to follow.	We are pleased that you find the paper to be mostly well written and have addressed your specific constructive suggestions for improvement below.

Abstract During the Covid-19 pandemic, there would be many people who are staying at home, and attempting to reduce contact with other people. Additional clarification is needed to identify these individuals. Clarification is needed in what is meant by ‘as much as possible’ in relation to individuals who are self-isolating. If keeping the concepts relating to the Safety I and II perspective in the abstract, these concepts should be briefly defined. If not, then these concepts should be removed from the abstract.	Thank you for this opportunity for clarification. We have clarified the inclusion criteria in the abstract as “People who have been advised to cocoon/shield or over the age of 70,’ as the terms cocoon/shield are linked to specific public health guidance for those experiencing listed medical vulnerabilities. We also agree it is best to remove the sensitising concept Safety I/II from the abstract, as above.
Introduction There are a number of terms in the introduction that should be defined more specifically to aid understanding. Clarification is needed about what is meant by ‘normal times.’ Similarly, an explanation is needed of what is meant by ‘lay involvement’ in medication practices, as well as of ‘medication performance’ and of ‘medication work.’	Thanks for this prompt to reflect on the terminology used. We have addressed each of these suggestions as follows: Normal times – We removed this term and specified the time periods in question. Lay involvement – page 4, line 22. Change “Evidence suggests lay involvement” to “Evidence suggests that contribution by patients, informal carers, family members (lay involvement’), We also added a clarification to the following sentence: “Examples of lay involvement include: patients and family members routinely take ...” Medication performance – We have removed this concept from the paper. Medication work – We have reviewed this and feel this term has been comprehensively introduced (page 4, line 34 onward). We hope that removing reference to the concept of medication performance now makes the concept of medication-related work clearer.
The authors stated that evidence exists identifying that medication taking, and medication work, is a highly personalised and complex phenomenon that may be contingent upon, and contextualised by, one’s situation, environment, processes and network. However, they provided little explanation of the complexities relating to influences in relation to what is already known from available evidence.	Thank you for this observation. While there is not room to comprehensively expand upon these complexities in this protocol paper, we have added the following to the manuscript: “Examples of such complexities include the medication user’s individualised routines and

	strategies, the nature of medication(s) involved, the range of people involved, or the individual's illness burden (11)."
It is not clear what the authors mean by 'other events of major public health concern,' since in the preceding part of the sentence, they referred to lack of details about medication safety during pandemics.	Thanks again, we have now removed 'major public health concerns' and focused on pandemics as follows: "Relatively little is known about medication safety or changes to household medication practices during a pandemic".
The authors stated that previous studies describing drug-related problems have been undertaken from the healthcare professionals' perspective, rather than from the person taking the medication. Does this statement	We agree with the reviewer's statement and have taken the opportunity to highlight that our point related to the DRP classification system, rather than the topic of DRPs.
refer during the time of a pandemic or outside of a pandemic event? There have been numerous studies undertaken of the patients' perspective in relation to drug-related problems that consider the broader concepts of household medication practices, which the authors have not described or critiqued.	We have changed this text to: "Previous studies have generally classified drug-related problems from a healthcare professional perspective, rather than that of the person taking the medication or their social network; drug-related problems are also generally identified in the context of medication review, rather than routine household settings (35)".
The aims and the specific objectives of the study appear appropriate.	Thank you for this positive feedback.
Methods and analysis It is stated that patient and public involvement partners will be involved in data analysis. Clarification is needed about whether these individuals were involved in the development phases of the project.	Thank you for the opportunity to clarify and highlight our reporting of this. We included this detail under the section titled "patient and public involvement" and have amended our wording slightly to make this more explicit. " PPI commenced during the development phase by partners validating our research aim and reviewing all participant-facing materials including recruitment materials, participant information leaflet, consent form and interview topic guide".
For inclusion criteria, clarification is needed about which adults comprise those individuals who experience a medical vulnerability as it could be assumed that many individuals could encompass this group. Similarly for the remaining two criteria, there are many adults who are on at least one long-term medication or who have assistance from another individual with medication management.	Thanks for the opportunity to clarify this. The terms "shield" and "cocoon" are specific terms used by the UK and Irish government public health departments and have specific listed medical vulnerabilities or conditions associated with them. We have inserted the references to the websites providing this information. We have also reworded our inclusion criteria slightly to make this clearer.

While the authors intend to use a convenience sampling process, they will develop a matrix to document participants' backgrounds to ensure focused recruitment of a diverse sample. It would be helpful to know about the types of individuals who utilise patient and carer advocacy groups, engagement with public and patient involvement, personal networks and social media. It is important potential participants comprise a wide array of characteristics to ensure adequate representation. It is encouraging to know that the authors will also seek out recruitment from charity groups.	Thanks for this. We have now included detail of the variables we will include in our matrix. We agree with the reviewer that it would be helpful to know about the types of individuals who utilise these groups and media and who participate in PPI. We will report on the study participant profile in our findings paper and we will reflect on any limitations associated with this. We are also mindful as recruitment progresses to make attempts to reach those who may not use the internet or social media.
	Thanks for the invitation to clarify this.

The actual process of the recruitment has not been described, and further information is needed about this activity. How will the authors obtain the potential participants' contact details in order to arrange an interview?	Prospective participants will be encouraged to make contact with a member of the research team, by telephone or email. Consistent with ethical, data protection and confidentiality regulations, we will not ask third parties to pass prospective participants' personal details to the research team. We have now clarified this in the manuscript.
For data analysis, clarification is needed about how sociotechnical assessment approaches using Safety I and II and Schafheutle et al.'s medication work framework will be applied.	Thank you. We have clarified that rather than apply Safety I/II approaches, we will use them as sensitising concepts, by which we mean: 'to explore the data, in which their application was used to inspire insights appropriate to the data rather than being used as a rigid coding framework.'
For research rigour, the authors stated that a proportion of interviews will be quality assured by another member of the research team for validation of coding and analysis. Further explanation is needed about this process.	Thank you, we have now revised this to be clearer. It now reads: "A proportion of interviews will be coded independently by at least two members of the research team, followed by a comparison of and discussion of coding as a means to quality assure the coding process"
Impact The authors seek to develop a person-centred medication practice safety framework for individuals staying at home and with reduced contact with others. From what is provided in the analysis section, which involves a constructivist-interpretive analysis approach, it is not clear how a person-centred medication practice safety framework will be developed. Further clarification is needed about the process of development of this framework and of the ways in which development of this framework will be influenced by other frameworks to be used during analysis.	Thank you for this. Upon reflection, we agree with this point and recognised that our choice of terminology was misleading. We have now revised our work to outline that we aim to generate best practice guidance from the patient/carer perspective and have removed references to 'a person-centred medication practice safety framework'.

Reviewer: 3 Reviewer Name: Dianne Goeman Institution and Country: University of Newcastle, Australia Please state any competing interests or state 'None declared': None declared	RESPONSES
This protocol paper describes a study that will explore the household medication practices of people over 70 yrs of age in the UK and the Republic of Ireland who are confined to home during the Covid-19 pandemic. The	Thank you. We are pleased that you find the paper to be well written and that you recommend publication. w

VERSION 2 – REVIEW

REVIEWER	DK Theo Raynor University of Leeds, UK I am academic advisor to Luto Research which develops, refines and tests health information materials. I am European co-chair of the Bayer 'Patient Safety Communication Advisory Panel'.
REVIEW RETURNED	10-Nov-2020

GENERAL COMMENTS	Thank you for your responses to my suggestions - I am happy to accept the revised manuscript.
---

REVIEWER	Elizabeth Manias Deakin University, Australia
REVIEW RETURNED	07-Nov-2020

GENERAL COMMENTS	The authors have now addressed all my concerns - thank you.
---